# Analysis of the Effects of Ninjin’yoeito on Physical Frailty in Mice

**DOI:** 10.3390/ijms231911183

**Published:** 2022-09-23

**Authors:** Shotaro Otsuka, Keita Fukumaru, Akira Tani, Seiya Takada, Kiyoshi Kikuchi, Kosuke Norimatsu, Ryoma Matsuzaki, Teruki Matsuoka, Harutoshi Sakakima, Yuji Omiya, Keita Mizuno, Yosuke Matsubara, Ikuro Maruyama

**Affiliations:** 1Department of Systems Biology in Thromboregulation, Kagoshima University Graduate School of Medical and Dental Science, 8-35-1 Sakuragaoka, Kagoshima 890-8520, Kagoshima, Japan; 2Department of Physical Therapy, School of Health Sciences, Faculty of Medicine, Kagoshima University, 8-35-1 Sakuragaoka, Kagoshima 890-8520, Kagoshima, Japan; 3Division of Brain Science, Department of Physiology, Kurume University School of Medicine, 67 Asahi-Machi, Kurume 830-0011, Fukuoka, Japan; 4Department of Neurosurgery, Kurume University School of Medicine, 67 Asahi-Machi, Kurume 830-0011, Fukuoka, Japan; 5Tsumura Kampo Research Laboratories, Tsumura & Co., 3586 Yoshiwara, Ami-Machi 300-1192, Ibaraki, Japan

**Keywords:** ninjin’yoeito, physical frailty, SAMP8, 8-hydroxy-2′-deoxyguanosine

## Abstract

Physical frailty is an aging-related clinical syndrome involving decreases in body weight, mobility, activity, and walking speed that occurs in individuals with sarcopenia and is accelerated by increased oxidative stress. Ninjin’yoeito, a traditional Japanese Kampo medicine, is used for treating conditions, including anemia and physical weakness. Here, we investigated whether ninjin’yoeito could improve physical frailty by controlling oxidative stress in the senescence-accelerated mouse prone 8 (SAMP8) model. First, SAMP8 mice were divided into two groups, ninjin’yoeito treated and untreated, with the former consuming a diet containing 3% ninjin’yoeito from 3 months of age. At 7 months of age, body weight, motor function, locomotor activity, and mean walking speed were measured. Subsequently, mice were euthanized and measured for muscle weight, 8-hydroxy-2′-deoxyguanosine levels in muscle and brain, and cleaved caspase-3 expression in brain. The results showed reductions in weight, locomotor function, locomotion, and average walking speed in the untreated group, which were significantly improved by ninjin’yoeito. Furthermore, 8-hydroxy-2′-deoxyguanosine levels were reduced in muscle and brain from ninjin’yoeito-treated mice, compared with the levels in untreated mice; cleaved caspase-3 expression was similarly reduced in brain from the treated mice, indicating reduced apoptosis. Our findings suggest that ninjin’yoeito inhibits sarcopenia-based physical frailty through its antioxidant effects.

## 1. Introduction

Aging commonly causes physiological changes, such as elevated levels of pro-inflammatory cytokines and enhanced mitochondrial dysfunction, as well as more frequent impairments in both tissue repair mechanisms and physical function. These changes are directly related to the risk of frailty in older people [1,2,3]. Frailty is a complex geriatric syndrome that involves age-associated declines in physiological reserve and function across multiple organ systems [4]. There are multiple types of frailty, including cognitive, physical, and social. Notably, physical frailty is defined as the presence of at least three of the following five criteria: unintentional body weight loss, fatigue, low physical activity, slow walking speed, and muscle weakness; the basis of these clinical manifestations is sarcopenia, a loss of skeletal muscle mass that leads to aging-related loss of muscle power and strength [5,6]. Because of crosstalk between the overall frail state and sarcopenia, muscle atrophy is regarded as the biological basis of physical frailty and is, therefore, referred to as physical frailty [1,5,7,8,9]. The muscular and nervous systems have established roles in the maintenance of physical function in older people. Most research concerning physical frailty has been focused on the role of the skeletomuscular system. However, there is also evidence that changes in the central nervous system may contribute or evolve in parallel to physical frailty [10]. Previous neuroimaging studies have shown that cerebrovascular damage, gray matter atrophy, and higher white matter hyperintensity volume (associated with reduced lower body function [11]) may contribute to the pathology of physical frailty [12,13,14,15].

Oxidative stress reportedly increases during the aging process and may play a role in the development of physical frailty [9]. Furthermore, oxidative stress might lead to the activation of apoptotic pathways associated with cellular damage, aberrations in the expression patterns of many transcription factors responsible for modulating protein synthesis and protein degradation, reductions in mitochondrial function, and impairments in repair mechanisms [16,17]. The brain and muscles are susceptible to increasing amounts of oxidative stress damage over time because they are responsible for most resting oxygen consumption [18,19,20]. A previous study revealed elevated levels of 8-hydroxy-2′-deoxyguanosine (8-OHdG), a marker of nucleic acid oxidation, in circulating lymphocyte DNA from patients with Alzheimer’s disease [21,22]. 8-OHdG is a major spontaneously oxidized derivative of 2′-deoxyguanosine and a biomarker of oxidative DNA damage. It is also expressed in skeletal muscles [23,24]. Finally, high serum 8-OHdG levels have been shown to be independently associated with physical frailty in older Chinese people [25].

The traditional Japanese medicine (Kampo medicine) ninjin’yoeito (NYT) has been approved as a prescription medication in Japan to improve recovery from various disorders, including anemia, fatigue, loss of appetite, anorexia, night sweats, and cold sensations in the extremities. NYT is typically prescribed as oral medication in Japan. NYT is a multicomponent drug that includes 12 crude herbs and plant products: rehmannia root, angelica root, atractylodes rhizome, Poria sclerotium, ginseng, cinnamon bark, polygala root, peony root, citrus unshiu peel, astragalus root, glycyrrhiza, and schisandra fruit [6,26,27]. Thus far, NYT has been shown to ameliorate muscle-related complications in cancer-bearing mice and chronic obstructive pulmonary disease model mice [28,29]. In addition, multiple constituents of NYT reportedly have antioxidant effects, including citrus unshiu peel [30], hesperidin [31], and Wolfiporia cocos (Schwein.) Ryvarden et Gilb. [32] showed that prepared NYT has also demonstrated antioxidant effects in vitro [33]. These effects of NYT may be clinically relevant because oxidative stress is considered fundamental to the pathophysiology of aging [34]. One key benefit of NYT is its antioxidant properties, as several of its constituents have such properties. DNA damage caused by oxidative stress activates the apoptotic pathway and promotes the expression of caspase-3, the final determinant of apoptosis [35]. From these reports, we hypothesize that the antioxidant effect of NYT may inhibit apoptosis via the inactivation of caspase-3 by reducing oxidative stress. However, to the best of our knowledge, few in vivo studies have clarified the effect of NYT on senescent physical frailty, despite its potential as an anti-aging drug.

The senescence-accelerated mouse prone 8 (SAMP8) model is a non-genetically modified mouse strain that exhibits an accelerated aging process. These mice demonstrate characteristics similar to the characteristics of aged humans, such as shortened lifespan, lordosis, hair loss, and reduced physical activity [36]. Moreover, they have been shown to exhibit typical features of skeletal muscle senescence, including high oxidative stress, greater declines in muscle mass and contractility, larger reductions in type II muscle fiber size, and slower muscle fiber contraction speed at a relatively young age, compared with senescence-resistant mouse 1 (SAMR1) mice at the same age [37,38]. SAMP8 mice have been used in many aging studies. Animal models of physical frailty that have been developed to date may not fully represent human physical frailty; for example, interleukin 10-knockout (IL-10KO) mice and superoxide dismutase 1-knockout (Sod1KO) mice are available, but the genes that cause the manifestations of these models of physical frailty may not contribute to physical frailty in humans. Moreover, IL-10KO mice and Sod1KO mice develop unique pathologies not observed in aged wild-type mice, such as chronic enterocolitis in IL-10KO mice and hepatocellular carcinoma in Sod1KO mice [39,40,41]. In addition, it has been reported that Sod1KO mice may not have sarcopenia [42]. The SAMP8 mice used in this study have been used extensively in aging research, especially in sarcopenia studies, while the IL-10KO and Sod1KO mouse models of frailty have not been used extensively in sarcopenia studies. Against this background, we hypothesized that SAMP8 could cause sarcopenia-based physical frailty and that using SAMP8 would allow for examination of the effects of NYT on physical frailty [37,43,44,45].

In this study, we investigated the effects of NYT on aging-related physical frailty by determining the levels of oxidative stress in muscle and brain tissues from SAMP8 mice. We hypothesized that oxidative-stress-induced damage to skeletal muscle and brain may contribute to reduced motor function and activity, on the basis of prior findings in other animal models [46,47,48,49]. Because reduced apoptosis has been observed [50,51,52] following treatment with components of NYT, we also investigated NYT-associated changes in cleaved caspase-3, a marker of apoptosis, in brain tissue from SAMP8 mice.

## 2. Results

### 2.1. SAMP8 Mice Exhibited Typical Features of Physical Frailty at 7 Months of Age

We first examined whether SAMP8 mice exhibit physical frailty during the aging process. No mice met the criteria for euthanasia during the experimental period and all mice completed the experimental period. The SAMR1 group showed an increase in body weight from 3 to 7 months of age. However, mice in the SAMP8 Control group (*n* = 8, 30.1 ± 1.1 g) showed significantly smaller body weight, compared with mice in the SAMR1 group (*n* = 8, 42.6 ± 1.3 g) at 7 months of age (*p* < 0.001; Figure 1A). Furthermore, walking time in the rotarod test was significantly shorter in the SAMP8 Control group (*n* = 8, 27 ± 4.4 s) than in the SAMR1 group (*n* = 8, 87.1 ± 6.7 s) (*p* < 0.001; Figure 1B). Grip strength was also significantly lower in the SAMP8 Control group (*n* = 8, 0.9 ± 0.1 N) than in the SAMR1 group (*n* = 8, 1.6 ± 0.1 N) (*p* < 0.001; Figure 1C). Distance traveled was significantly reduced in the SAMP8 Control group (*n* = 8, 12,246.8 ± 690.0 cm) compared with that in the SAMR1 group (*n* = 8, 15,616.4 ± 999.1 cm) (*p* = 0.025; Figure 1D). Furthermore, the mean speed was significantly reduced in the SAMP8 Control group (*n* = 8, 3.4 ± 0.2 cm/s) compared with that in the SAMR1 group (*n* = 8, 4.3 ± 0.3 cm/s) (*p* = 0.024; Figure 1E).

### 2.2. NYT Reduced Manifestations of Physical Frailty in SAMP8 Mice

We next evaluated whether physical frailty could be reduced by NYT administration. At 7 months of age, mice in the SAMP8 NYT group (*n* = 8, 36.1 ± 2.1 g) exhibited greater body weight than mice in the SAMP8 Control group (all values for SAMP8 Control mice are provided above; *p* = 0.011; Figure 1A). Walking time in the rotarod test was significantly greater in the SAMP8 NYT group (*n* = 8, 46.4 ± 4.4 s) than in the SAMP8 Control group (*p* = 0.02; Figure 1B). Grip strength was also significantly greater in the SAMP8 NYT group (*n* = 8, 1.5 ± 0.1 N) than in the SAMP8 Control group (*p* < 0.001; Figure 1C). Both distance traveled and mean speed were greater in the SAMP8 NYT group (*n* = 8, 18,314.1 ± 1194.9 cm, 5.1 ± 0.3 cm/s) than in the SAMP8 Control group (both *p* < 0.001; Figure 1D,E). These results indicated that NYT improves body weight loss, motor function decline, and locomotor activity reduction.

### 2.3. NYT Suppressed Reductions in Plantar Muscle Fiber Size and Muscle Cross-Sectional Area in SAMP8 Mice

At 7 months of age, none of the mice exhibited aging-related soleus muscle atrophy (*n* = 8 per group, SAMR1 group 0.025 ± 0.001%, SAMP8 Control group 0.022 ± 0.003%, SAMP8 NYT group 0.022 ± 0.003%). However, both SAMP8 groups (*n* = 8 per group, SAMP8 Control group 0.039 ± 0.003%, SAMP8 NYT group 0.04 ± 0.022%) exhibited significantly smaller plantar muscle weight to body weight ratios, compared with the ratio in the SAMR1 group (*n* = 8, 0.051 ± 0.002%, both *p* < 0.001; Figure 2A). The type II fibers of plantar muscles were atrophied in both SAMP8 groups, compared with those in the SAMR1 group, indicating the presence of sarcopenia. The plantar muscle weight to body weight ratio did not significantly differ between the SAMP8 NYT and SAMP8 Control groups (Figure 2A). Furthermore, the plantar muscle cross-sectional area was significantly smaller in the SAMP8 Control group (*n* = 5, 869.2 ± 37.8 μm^2^) than in the SAMR1 group (*n* = 5, 1254.0 ± 190.0 μm^2^, *p* = 0.048; Figure 2B,C). The SAMP8 NYT group (*n* = 5, 1373.8 ± 91.8 μm^2^) showed significantly greater muscle cross-sectional area compared with the SAMP8 Control group (*p* = 0.014, Figure 2B,C). These results indicated that NYT improves plantar muscle age-dependent atrophy.

In this study, we used 8-OHdG nuclear staining as an indicator of oxidative DNA damage in muscle tissue. The SAMP8 Control group (*n* = 5, 59.7 ± 6.5%) had significantly greater 8-OHdG immunoreactivity than the SAMR1 group (*n* = 5, 14.3 ± 1.1%, *p* < 0.001; Figure 3A,B). The number of 8-OHdG-positive nuclei was significantly smaller in the SAMP8 NYT group (*n* = 5, 35.1 ± 4.3%) than in the SAMP8 Control group (*p* = 0.002; Figure 3A,B). We also assessed whether aging-related oxidative stress is present in cortical brain tissues. Aging led to greater numbers of 8-OHdG-positive neurons in cortical regions of the brain; in particular, the SAMP8 Control group (*n* = 5, 39.6 ± 3.1%) had significantly more 8-OHdG-positive neurons in cortical brain tissues than the SAMR1 group (*n* = 5, 15.8 ± 3.9%, *p* < 0.001; Figure 4A,B). Notably, there were significantly fewer 8-OHdG-positive neurons in the SAMP8 NYT group (*n* = 5, 25.1 ± 2.3%) than in the SAMP8 Control group (*p* = 0.006; Figure 4A,B).

### 2.4. NYT Reduced Physical Frailty-Related Apoptosis in Brain Tissues

Because enhanced oxidative stress induces brain apoptosis [50,51,52], we used Western blotting to measure the protein expression of cleaved caspase-3, a marker of apoptosis; this expression was significantly greater in the SAMP8 Control group (*n* = 4, 6.4 ± 0.4) than in the SAMR1 group (*n* = 4, 1.4 ± 0.3, *p* < 0.001; Figure 5A,B). Similarly, we found significantly greater cleaved caspase-3 immunoreactivity in cortical brain tissues from the SAMP8 Control group (*n* = 5, 6.4 ± 0.3%) than from the SAMR1 group (*n* = 5, 2.3 ± 0.4%, *p* < 0.001; Figure 5C,D). Furthermore, the protein expression of cleaved caspase-3 was significantly lower in the SAMP8 NYT group (*n* = 4, 3.7 ± 0.7) than in the SAMP8 Control group (*p* = 0.03; Figure 5A,B). In addition, the SAMP8 NYT group (*n* = 5, 3.9 ± 0.4%) exhibited significantly diminished cleaved caspase-3 immunoreactivity in cortical brain tissues compared with the SAMP8 Control group (*p* = 0.001; Figure 5C,D).

## 3. Discussion

In this study, we presumed that SAMP8 mice would exhibit sarcopenia-related physical frailty symptoms. Our findings confirmed that SAMP8 mice developed physical frailty by 7 months of age. Furthermore, we investigated the effects of NYT on aging-related physical frailty and the levels of oxidative stress in muscle and brain tissues from SAMP8 mice. We demonstrated that NYT improved physical frailty with respect to body weight, motor function, locomotor activity, and muscle atrophy; these effects were presumably mediated through reductions in oxidative stress in both muscle and brain tissues. Finally, we examined whether the antioxidant stress effects of NYT could inhibit apoptosis induced by oxidative stress and confirmed that NYT administration inhibited cleaved caspase-3.

SAM animals are classified as either SAMR or SAMP; the SAMP8 model is one of nine types of SAMP mice [53]. A previous study of physical frailty in aged mice focused on body weight, grip strength, motor function, locomotor activity, and walking speed [54,55]. Here, we used similar measurements, as well as skeletal muscle atrophy and muscle cross-sectional area. At 7 months of age, we found that SAMP8 mice showed significant body weight loss, as well as reduced motor function, activity, and gait speed, compared with SAMR1 mice. Additionally, we found plantar muscle atrophy and reduced cross-sectional area in type II plantar muscle fibers. Cobley et al. reported that aging causes a reduction in muscle type II fibers [19]. Furthermore, SAMP8 mice have been shown to exhibit atrophy of skeletal muscles and other tissues at the age of 7 months, resulting in sarcopenia [45]. The present study showed that SAMP8 mice demonstrate sarcopenia-related physical frailty at 7 months of age, indicating that these mice constitute a suitable in vivo model for the study of physical frailty.

Sarcopenia plays a fundamental part in physical frailty [56]. Sarcopenia in animal models is defined using a combination of muscle atrophy, decreased muscle cross-sectional area, decreased muscle function, and muscle weakness [57]. The 2018 European Working Group on Sarcopenia in Older People 2 consensus regarded weak muscle strength as an important feature of sarcopenia; currently, muscle strength comprises the most reliable measurement index [58]. In this study, we found no differences in muscle mass; however, we observed improvements in muscle cross-sectional area and the results of grip strength test and rotarod test in SAMP8 mice that were fed NYT. Therefore, our investigation showed that NYT has the potential to improve sarcopenia.

NYT is a Japanese herbal medicine used to facilitate recovery from disease or postoperative fatigue [29]. Because NYT consists of 12 herbs and roots, it has various effects. Ginseng, a component in NYT, has been shown to promote neurogenesis in rat brain tissue following ischemia/reperfusion injury [52]. Another component, astragalus root, enhances insulin sensitivity by increasing adiponectin, especially its potent high-molecular-weight form [59]. Tenuigenin, the active ingredient in polygala root, promotes the proliferation and differentiation of hippocampal neural stem cells [60]. In this study, we found that treatment with NYT led to amelioration of body weight loss, hypomobility, and diminished locomotor activity in SAMP8 mice at 7 months of age. Furthermore, mice treated with NYT exhibited greater muscle cross-sectional area. NYT has been shown to improve appetite loss and body weight loss in cisplatin-treated mice by activating neuropeptide Y [61], which is involved in controlling appetite. Previous studies have shown that NYT and citrus unshiu peel could increase food intake by activating orexigenic orexin 1 receptor-expressing neurons in the hypothalamus [62]. We, therefore, suspect that the appetite-stimulating effect of NYT prevents body weight loss due to physical frailty.

Additionally, we focused on the reduction in oxidative stress as a possible mechanism underlying the inhibitory effect of NYT on physical frailty. 8-OHdG, a marker of oxidative-stress-induced DNA damage, is associated with physical frailty and is enhanced in muscle and brain tissues in older people [21,22,23,25]. Previous studies have indicated that greater 8-OHdG levels imply heightened oxidative-stress-induced damage during the aging process. Our results showed that 8-OHdG levels were elevated in plantar muscles and cortical regions of the brain in SAMP8 mice with aging-related physical frailty. The maintenance of both brain and skeletal muscle functions is essential for the preservation of locomotor activity and motor function. Oxidative stress causes muscle atrophy through increased muscle proteolysis [63,64]. Furthermore, oxidative stress can cause brain damage and corresponding functional decline [65,66]. Notably, older people with physical frailty show lower total brain and gray matter volumes, compared with healthy older people [14,67].

SAMP8 is a mouse model of aging that is susceptible to oxidative-stress-induced damage [68]; here, we presumed that oxidative-stress-induced damage to skeletal muscle and brain would contribute to reduced motor function and activity. Our results showed that NYT suppressed the levels of 8-OHdG in brain and muscle, ameliorating the reduction in muscle cross-sectional area and the enhancement of brain apoptosis; these findings were consistent with previous research concerning the reduction in oxidative stress [31,32,33] and reduction in apoptosis [50,51,52] following treatment with components of NYT. The antioxidant effect of NYT on the brain is mediated by its constituents, including cinnamon bark, polygala root, and citrus unshiu peel [69]. A recent report showed that cinnamaldehyde (present in cinnamon bark) has potent neuroprotective effects against oxidative stress and apoptosis induced by glutamate in the rat pheochromocytoma PC12 cell line (a neuron model) [70]. In addition, Sedighi et al. reported that an extract of cinnamon bark prepared with ethanol could protect heart tissue against ischemia/reperfusion injury through an antioxidant mechanism [71]. Tenuigenin (present in polygala root) has been reported to exhibit antiapoptotic and antioxidative activities in hippocampal neurons because of its ability to scavenge intracellular reactive oxygen species in cultured hippocampal neurons [72]. Hesperidin (present in citrus unshiu peel) ameliorates cognitive dysfunction, oxidative stress, and apoptosis in an aluminum chloride-induced rat model of Alzheimer’s disease [31]. Many previous studies have shown that the antioxidant properties of the constituents in NYT are among its key benefits. DNA damage caused by oxidative stress activates the apoptotic pathway and promotes the expression of caspase-3, the final determinant of apoptosis [35]. NYT has been reported to increase the expression of NGF [73,74] and it has been reported that the antioxidant properties of NGF can inactivate caspase-3, which may, in turn, lead to apoptosis inhibition. Furthermore, cinnamon bark, a component in NYT, induces NF-E2-related factor 2 (Nrf2) [75], a potent antioxidant, and previous studies reported that Nrf2 can protect aging skeletal muscle from oxidative stress [76]. Factors, such as NGF and Nrf2, may be associated with improvements in physical frailty by suppressing oxidative stress in NYT in this experiment. We hypothesize that these components in NYT reduced oxidative stress in aging brain tissue in SAMP8 mice, thereby suppressing neuronal cell death. Taken together, our findings suggest that NYT ameliorates physical frailty by inhibiting muscle and brain damage associated with aging.

Although efforts were made to reduce standard biases associated with animal experiments and tissue analysis, this study had some limitations. First, SAMP8 mice are vulnerable to oxidative stress and are at risk of excessive muscle and brain damage, compared with normal aging [68,77,78]; accordingly, the effects of NYT should be confirmed in other animal models. Second, we did not measure food intake to evaluate the main effect of NYT [61], which involves appetite enhancement. Therefore, the mechanism that improved body weight loss due to physical frailty remains unknown. The possibility that the appetite-stimulating effect of NYT may affect weight loss and muscle atrophy associated with aging should be investigated. The dosage of the NYT-containing diet in the present experiment was 5 g/mouse/day; in a previous study using the same NYT-containing diet [29], the mean intake per C57BL/6 mouse was 3.39 ± 0.027 g, suggesting that the dosage in the present experimental diet was sufficient during NYT administration. Moreover, a previous study regarding food intake showed that 5 g/day for 12 weeks was sufficient in SAMP8 mice [79], implying that food restriction did not influence our findings. Third, the mechanism of antioxidant action by NYT needs to be examined in more detail and we believe that the pharmacological actions of NYT might be clarified by evaluating the expression of antioxidants, such as Nrf2. Fourth, although we focused on oxidative stress as a potential factor underlying the aging process, enhanced inflammation is also an important factor involved in aging; it has been associated with greater mortality and the onset of physical frailty in older people [80]. Aging-related enhancements in inflammation may also influence skeletal muscle atrophy and brain damage [19,81]. Fifth, the dose-dependent effect of NYT used in this study is unknown. In vitro studies using isolated rat hepatocytes have reported a volume-dependent increase in the antioxidant effect of NYT [82]; therefore, in the present experiments, a larger volume of NYT may provide better results. Finally, the purpose of the study was to investigate the effects of NYT on frailty in mice with accelerated aging, rather than to investigate the effects of NYT on non-frail mice that do not undergo accelerated aging. Therefore, no experiments were conducted to administer NYT to SAMR1 mice and the effects of NYT on non-frail muscle and brain should be studied in future investigations. Further studies are needed to provide direct evidence of a causal relationship between NYT-induced improvements in physical frailty and cellular mechanisms; however, the present study suggests that the antioxidant stress effects of NYT may be an effective intervention for physical frailty in older adults and individuals with muscle weakness after surgery.

## 4. Materials and Methods

### 4.1. Animals

Sixteen 3-month-old male SAMP8 mice (body weight: 29.1 ± 0.9 g) and eight age-matched male SAMR1 mice (31.6 ± 1.6 g) were obtained from Japan SLC Animal Supply (Hamamatsu, Shizuoka, Japan). The SAMR1 mice were used as negative controls for the effects of aging in this study. No specific criteria were used to select the animals included in this study; the numbers of mice were determined in accordance with the sample sizes used in previous studies [29,37]. As described in the Introduction, SAMP8 mice constitute the ideal model mouse for this study. Mice were housed in a temperature-controlled (23.0 ± 1.0 °C) cage with a 12 h light/dark cycle and free access to food and water. To minimize potential confounders, only SAMP8 and SAMR1 mice were kept in a single room in the experimental animal facility, in a specific pathogen-free environment; up to three mice were housed in each plastic cage. The bedding material was composed of fine and soft wood chips. To maintain a rich environment, the breeding cages were changed at 2-day intervals. In this study, animal welfare considerations included all possible efforts to minimize suffering and distress, as well as the use of anesthetics (described in the Euthanasia subsection) and controlled housing conditions (described earlier in this paragraph). In our institution, experiments involving animal stress need to have a limited duration. However, the Ethics Board of Kagoshima University determined that no special measures were needed in this study because the pain was minor (e.g., a short period of holding, restraint, and injection). Researchers involved in all animal experiments in this study received customized training (in animal care and handling) designed by the Institute of Laboratory Animal Sciences of Kagoshima University, prior to performing any experiments. The experimental protocol was approved by the Ethics Board of the Institute of Laboratory Animal Sciences of Kagoshima University (approval number: MD18056).

### 4.2. NYT Preparation

The 16 SAMP8 mice in this experiment were randomly divided into two groups. The SAMP8 NYT group (*n* = 8) was fed a special diet containing 3% NYT (ninjin’yoeito TJ-108, lot no. 362113100 extract powder; Tsumura & Co., Tokyo, Japan) within the food pellets (Table 1). NYT TJ-108 was provided as grayish-brown granules, with characteristic stringent odor and sweet taste. The composition of NYT TJ-108 has been identified in previous studies [83]. Meanwhile, the SAMP8 Control group (*n* = 8) was fed a diet without NYT (KBTO190189; Oriental Yeast Co., Ltd., Tokyo, Japan); the SAMR1 mice were also fed a diet without NYT (KBTO190189). Mice in all three groups were fed a diet of 5 g/day per animal, in accordance with the findings in a previous study where each mouse’s mean food intake was 3.5–4.5 g/day for 12 weeks [79]. In a previous study, 3% NYT was used for treatment of experimental animals; therefore, 3% NYT was used in this study [29]. Mice were housed and fed in this manner until 7 months of age; they were then subjected to testing, euthanasia, and tissue-harvesting procedures by experimenters who had no knowledge of the grouping allocations. The duration of the experiment was 4 months, beginning at 3 months of age. Throughout treatment with NYT, daily assessments of overall health status were performed. Humane endpoints meriting euthanasia were significant weight loss (weight loss of ≥20% based on peak weight), the inability to ingest food, and a lack of mobility within the cage. If an animal met one or more endpoint criteria, euthanasia was required within 2 h. These endpoints were not reached during the study.

### 4.3. Motor Function Test

Motor function was examined in all mice using a motorized rotarod test (MK-670; Muromachi Kikai Co., Ltd., Tokyo, Japan) when the mice reached 7 months of age. Each mouse was placed on the rotarod cylinder and the duration that the mouse remained on it was measured. The rotation speed was increased from 0 to 40 rpm in increments of 4.0 rpm at intervals of 6 s. The trial ended if the animal fell into the cylinder. Each animal underwent two trials; the mean latency (s) was used in the analysis. Motor function tests were performed from 10:00 a.m. to 1:00 p.m.

### 4.4. Muscle Strength Grip-Strength Test

For the sarcopenia mouse model, the main assessment involves non-invasive measurement of forelimb gripping force, which can measure maximum forelimb contraction force during autonomous activity [84]. Muscle function has also been assessed using the rotarod method. To measure muscle strength in all mice, a commercially available Grip Strength Meter (MK-380Si; Muromachi Kikai Co., Ltd., Tokyo, Japan) was used when the mice reached 7 months of age. Each mouse was placed on the grid and was pulled by its tail with increasing force until the grip was broken. The maximum amount of force exerted (N) was recorded. This was repeated two times and the mean reading was calculated. Muscle strength grip-strength tests were performed from 10:00 a.m. to 1:00 p.m.

### 4.5. Behavioral Test

To minimize potential confounders, behavioral tests were performed on separate days from the motor function assessments. The locomotor activity of all mice was measured using an open-field test when the mice reached 7 months of age. An open-field apparatus (55 cm × 60 cm × 40 cm) was placed in a quiet environment and wiped clean with a 75% ethanol solution between tests. The mice were allowed to become acclimated to the center of the open field for 5 min and spontaneous activities were recorded for 1 h using a video camera (Logicool HD Pro Webcam C920r; Logicool Co., Ltd., Lausanne, Switzerland) mounted above the open field. The locomotor distance and mean speed were measured using the SMART version 3.0 video camera system (Panlab, Barcelona, Spain). Behavioral tests were performed from 10:00 a.m. to 5:00 p.m.

### 4.6. Euthanasia

Twenty-four male mice (16 SAMP8 and 8 SAMR1) were used in this study. All mice were able to complete motor, strength, and behavioral tests at 7 months of age (i.e., the end of the experimental period). Subsequently, all mice were deeply anesthetized via intraperitoneal injection of pentobarbital sodium (100 mg/kg) and transcranially perfused with physiological saline. This overdose of sodium pentobarbital is widely recommended in the literature for the purpose of euthanasia. They were then decapitated and the plantar muscle, soleus muscle, and brain tissues were collected. After tissue collection, muscle weights were measured.

### 4.7. Tissue Preparation

Muscle and brain tissues were fixed in 4% paraformaldehyde in 0.1 M phosphate buffer (pH 7.4) at 4 °C overnight. After fixation, the tissues were processed for histology and immunohistochemistry by dehydration and embedding in paraffin. The paraffin-embedded coronal muscle and brain sections were sliced at a thickness of 4 µm. Transverse sections were stained with hematoxylin and eosin (HE) to observe histological changes.

### 4.8. Assessment of Cross-Sectional Area

The HE-stained plantar myofibers were imaged at 10× magnification using a microscope and a digital camera. In accordance with a previously described method [37], the cross-sectional area of the myofibers was calculated from a minimum of 50 myofibers per animal using ImageJ software, version 1.46r (National Institutes of Health, Bethesda, MD, USA). The assessment of cross-sectional area was performed on five animals per group because suitable tissue sections were not available for the remaining three animals in all groups.

### 4.9. Histology and Immunohistochemistry

Tissues were probed with the following antibodies: rabbit polyclonal anti-cleaved caspase-3 (a marker of apoptotic activity) (Abcam, Cambridge, MA, USA; cat. no. ab2302), rabbit polyclonal anti-8-OHdG (a marker of oxidative stress) (Bioss Antibodies, Woburn, MA, USA; cat. no. bs-1278R), and mouse monoclonal anti-NeuN (a marker of neurons) (Abcam; cat. no. ab104224).

All tissues were subjected to deparaffinization and rehydration, followed by inhibition of endogenous peroxidase activity using methanol containing 3.0% hydrogen peroxide for 10 min. If necessary, antigen activation was then performed in accordance with the manufacturer’s protocol. The sections were then rinsed three times (5 min each) with phosphate-buffered saline (PBS, pH 7.6) and blocked with 10% skim milk in PBS for 20 min. Brain sections were individually incubated at 4 °C overnight with rabbit anti-cleaved caspase-3 antibody (1:200). The sections were then washed in PBS (three times for 5 min each) and incubated for 60 min with goat anti-rabbit IgG antibody, which had been conjugated to a peroxidase-labeled dextran polymer (EnVision; Dako, Carpinteria, CA, USA; cat. no. K4003). Finally, the sections were rinsed with PBS (three times for 5 min each) and immunoreactivity was visualized by diaminobenzidine staining. Co-staining was performed using sections of plantar muscle and brain tissues. Anti-mouse NeuN (1:200) and anti-rabbit 8-OHdG (1:200) immunoreactivities were examined by immunofluorescence staining in brain tissues. In plantar muscle, anti-rabbit 8-OHdG (1:200) immunoreactivity and 4′,6-diamino-2-phenylindole staining were examined by immunofluorescence staining. After overnight incubation with the above primary antibodies and PBS wash steps (three times for 5 min each) as described above, the sections were incubated for 60 min with both Alexa Fluor 488-conjugated goat anti-rabbit IgG antibody (1:100; Abcam; cat. no. ab150077) and Alexa Fluor 555-conjugated goat anti-mouse IgG antibody (1:100; Abcam; cat. no. ab150114). The sections were then washed with PBS (three times for 5 min each) and counterstained with 4′,6-diamino-2-phenylindole (1:500; Dojindo Laboratories, Kumamoto, Japan; cat. no. 340-07971) for 10 min. Finally, the sections were mounted with an aqueous mounting medium and observed with a fluorescence microscope (EVOS f1; AMG, Mill Creek, Snohomish County, WA, USA).

### 4.10. Quantitative Analysis of Immunostained Sections

Immunostained sections of plantar muscle were imaged at 20× magnification using a microscope and camera (EVOS f1; AMG). The number of 8-OHdG-positive nuclei was counted and expressed as a percent relative to the total number of nuclei in all mice (*n* = 5 for each group). The percentage of 8-OHdG for plantar muscle was calculated from two random, non-overlapping image fields (Online Resource Appendix A), in accordance with a previous study All quantitative analyses were performed using ImageJ software, version 1.46r (National Institutes of Health), by two experimenters (KF and AT) who were blinded to the group allocations. In addition, images of co-stained 8-OHdG and NeuN immunofluorescence in cortical brain tissues were captured at 20× magnification using a fluorescence microscope and a camera (EVOS f1; AMG). Brain tissues were measured in two locations (Online Resource Appendix A), in accordance with previous studies [48]. The number of NeuN and 8-OHdG double-positive neurons was counted in the motor cortex (0.74 mm^2^) in all mice (*n* = 5 for each group) and expressed as a percent relative to the total number of NeuN-positive neurons. The quantitative analysis of each immunolabeled area was performed using the ImageJ software described above, by two experimenters (KF and AT) who were blinded to the group allocations.

### 4.11. Western Blotting

Western blotting was performed to detect the protein levels in brain tissues from 7-month-old SAMP8 and SAMR1 mice (*n* = 4 for each group). The brain tissue was placed on ice and homogenized in T-Per reagent (Pierce Biotechnology, Rockford, IL, USA; cat. no. 78510). Approximately 10 μg of protein per sample was loaded in a 4–20% mini-protean precast gel (Bio-Rad, Hercules, CA, USA) and transferred to a polyvinylidene fluoride or nitrocellulose membrane. After the membrane had been blocked with Tris-buffered saline plus Tween 20 containing 5% skim milk for 1 h at room temperature, it was incubated with a primary antibody (described below) overnight at 4 °C and then with a secondary horseradish peroxidase-labeled antibody (goat anti-rabbit IgG H&L (1:4000; Abcam; cat. no. ab97051) or goat anti-mouse IgG H&L (1:5000; Abcam; cat. no. ab6789)) for 1 h at room temperature. Detection was performed using Immobilon chemiluminescent horseradish peroxidase substrate (Millipore). The following primary antibodies were used for Western blotting: rabbit polyclonal anti-cleaved caspase-3 (1:400; Abcam; cat. no. ab2302) and mouse monoclonal anti-α-tubulin (1:2000; Proteintech Group, Inc., Rosemont, Cook County, IL, USA; cat. no. 66031-1-lg). The protein bands were visualized with a chemiluminescence system (WSE-6100 Lumino Graph I; Atto, Tokyo, Japan) and measured using ImageJ software, version 1.46r (National Institutes of Health). Target band intensities were normalized to the intensities of α-tubulin bands from the same samples.

### 4.12. Statistical Analysis

Statistical analyses were performed with parametric tests, following confirmation that the data exhibited a normal distribution (determined using the Shapiro–Wilk test). Specifically, body weight, walking time, grip strength, behavioral test distance, and mean speed, as well as the proportions of cleaved caspase-3-positive areas, 8-OHdG-positive nuclei, and 8-OhdG-positive neurons, were analyzed by one-way ANOVA, followed by post hoc Fisher’s least significant difference tests. A *p*-value of <0.05 was considered statistically significant. Data are expressed as the mean ± standard error. All data were analyzed using IBM SPSS Statistics, version 26 (IBM Corp., Armonk, NY, USA). No data from any assays were excluded from the analysis for any reason.

## 5. Conclusions

Our findings indicated that SAMP8 mice showed body weight loss; reduced motor function, locomotor activity, and mean walking speed; muscle atrophy; and diminished type II muscle fiber cross-sectional area at the age of 7 months. These results suggest that SAMP8 mice can serve as a model of physical frailty based on sarcopenia; importantly, these mice may help researchers to avoid the limitations of available models (e.g., IL-10KO mice and Sod1KO mice). Treatment with NYT led to reduced oxidative stress in aged muscle and brain in SAMP8 mice. Overall, our findings suggest that NYT inhibits sarcopenia-based physical frailty through its antioxidant effects.

## Figures and Tables

**Figure 1 ijms-23-11183-f001:**
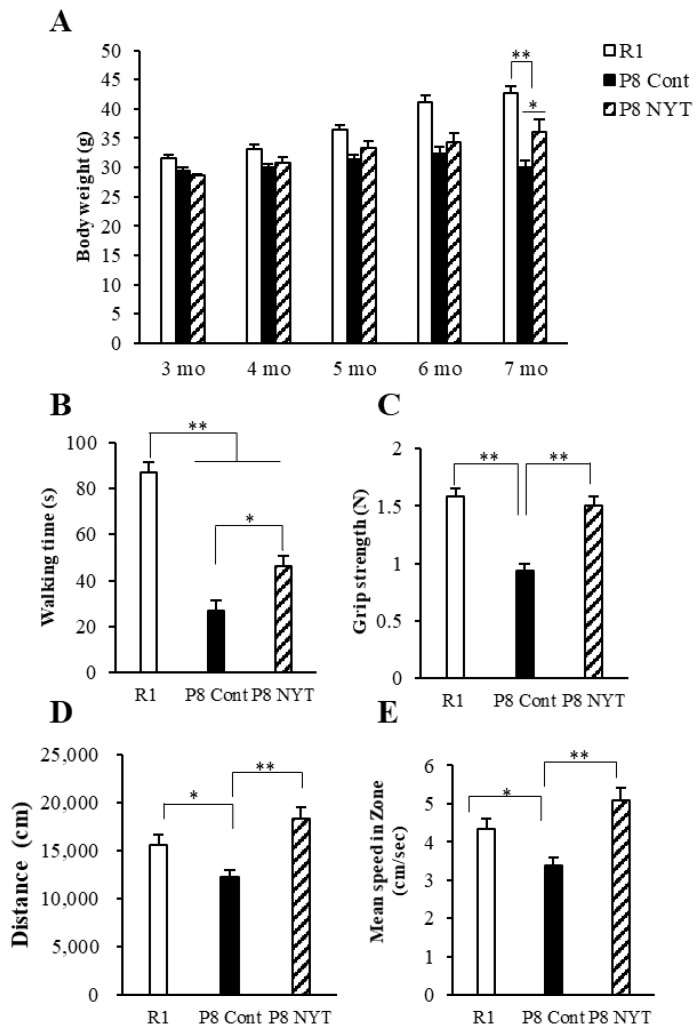
Effects of NYT on aging-related body weight reduction, motor dysfunction, and behavioral decline in SAMP8 mice. Body weight (**A**), rotarod test (**B**), grip strength test (**C**), and open-field test results (**D**,**E**) are shown. Body weight, motor function, distance traveled, and mean speed significantly declined with age. The SAMP8 NYT group (“P8 NYT” in the figure, *n* = 8) had significantly better body weight reduction, motor dysfunction, and behavioral decline at 7 months of age, compared with the SAMP8 Control group (“P8 Cont” in the figure, *n* = 8) (**A**–**E**). SAMR1 negative control mice (*n* = 8) are designated as “R1” in the figure. Data are presented as mean ± standard error. * *p* < 0.05, ** *p* < 0.01.

**Figure 2 ijms-23-11183-f002:**
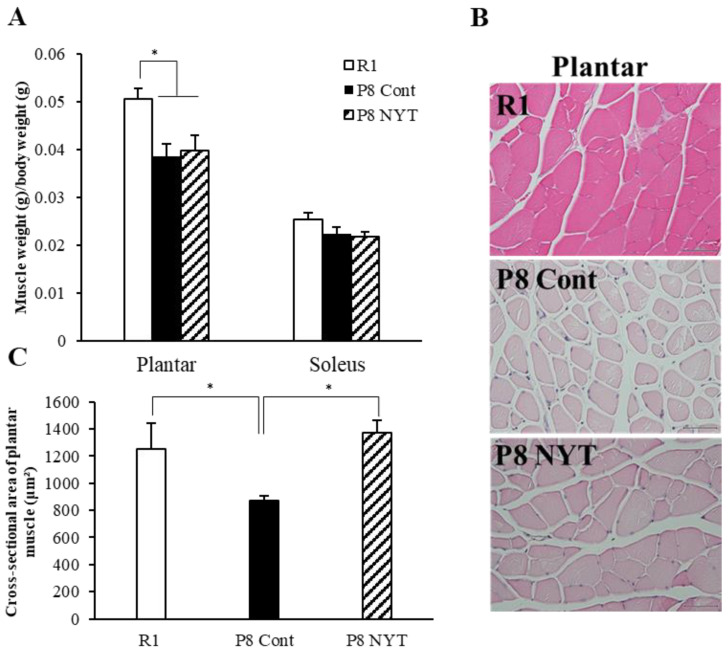
NYT suppressed reductions in plantar muscle fiber size and muscle cross-sectional area in SAMP8 mice. Muscle weight to body weight ratios (**A**), HE-stained plantar muscle tissues (**B**), and cross-sectional areas of type II fibers of plantar muscles (**C**) are shown. Both groups of SAMP8 mice (*n* = 8 per group) showed significantly fewer type II fibers of plantar muscles, compared with the SAMR1 group (“R1” in the figure, *n* = 8) at 7 months of age. The type I fibers of soleus muscles showed no aging-related changes. The cross-sectional area of plantar muscle was significantly greater in the SAMP8 NYT group (“P8 NYT” in the figure) than in the SAMP8 Control group (“P8 Cont” in the figure). Data are presented as mean ± standard error. * *p* < 0.05,; scale bars in all panels = 50 µm. In (**C**), *n* = 5 for all groups.

**Figure 3 ijms-23-11183-f003:**
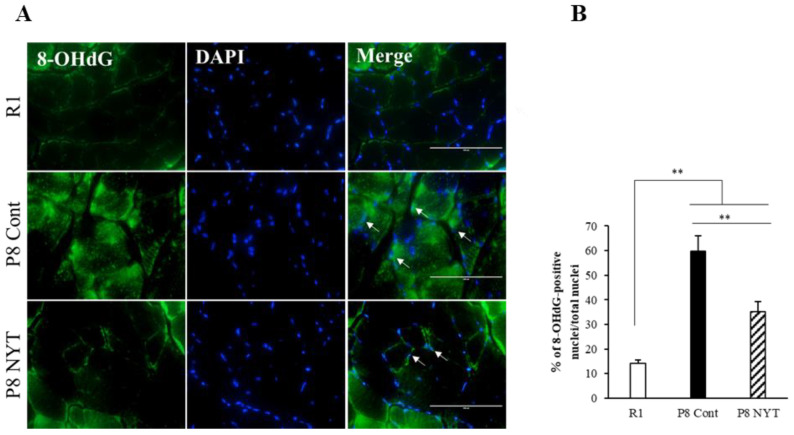
NYT reduced the aging-related increase in oxidative stress within plantar muscle. There was significantly more 8-OHdG co-staining in mice that exhibited aging. The level of 8-OHdG was significantly lower in the SAMP8 NYT group (“P8 NYT” in the figure, *n* = 5) than in the SAMP8 Control group (“P8 Cont” in the figure, *n* = 5) (**A**,**B**). White arrows indicate positive cells. SAMR1 negative control mice are designated as “R1” in the figure, *n* = 5. Data are presented as mean ± standard error., ** *p* < 0.01; scale bars = 100 µm (**A**).

**Figure 4 ijms-23-11183-f004:**
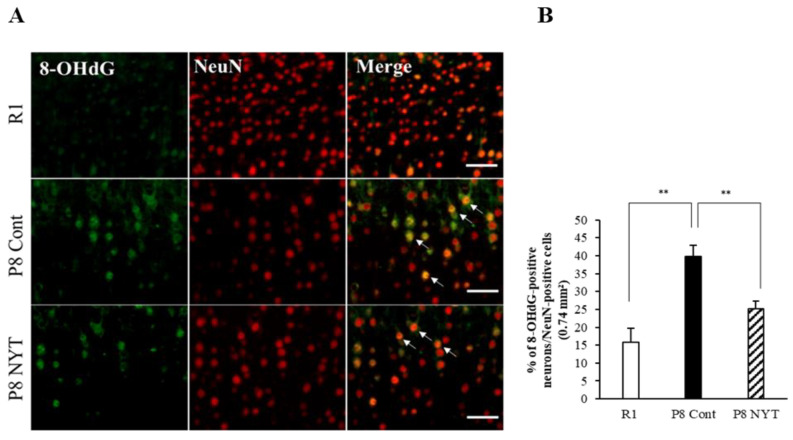
NYT reduced the aging-related increase in oxidative stress within cortical brain tissues. There was significantly more NeuN co-staining with 8-OHdG in mice that exhibited aging. The level of 8-OHdG was significantly lower in the SAMP8 NYT group (“P8 NYT” in the figure, *n* = 5) than in the SAMP8 Control group (“P8 Cont” in the figure, *n* = 5) (**A**,**B**). White arrows indicate positive cells. SAMR1 negative control mice (*n* = 5) are designated as “R1” in the figure. Data are presented as mean ± standard error., ** *p* < 0.01; scale bars = 100 µm (**A**).

**Figure 5 ijms-23-11183-f005:**
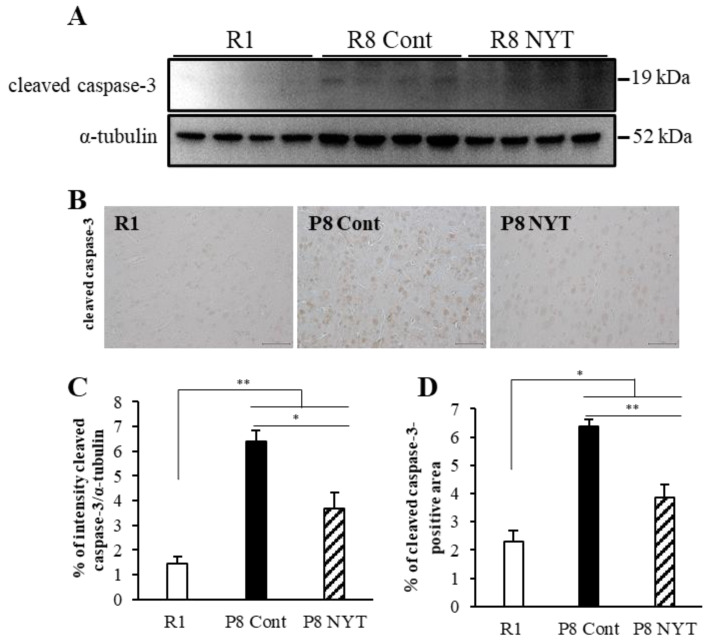
NYT inhibits aging-related apoptosis in brain tissues. Western blotting analysis of cleaved caspase-3 expression in brain tissues (**A**). Photomicrographs of cleaved caspase-3 immunoreactivity (**B**). Cleaved caspase-3 expression was significantly greater in mice that exhibited aging. Cleaved caspase-3 expression was significantly lower in the SAMP8 NYT group (“P8 NYT” in the figure) than in the SAMP8 Control group (“P8 Cont” in the figure) (**C**,**D**). SAMR1 negative control mice are designated as “R1” in the figure. Data are presented as mean ± standard error. * *p* < 0.05, ** *p* < 0.01; scale bars = 50 µm (**B**). In (**A**), *n* = 4 for all groups; in (**B**), *n* = 5 for all groups.

**Table 1 ijms-23-11183-t001:** Ninjin’yoeito composition.

Latin Name	Original Plant Source and Medicinal Part of Crude Drug	Amount (g)
REHMANNIAE RADIX	The roots of Rehmannia glutinosa Liboschitz var. Purpurea Makino or Rehmannia glutinosa Liboschitz	4.0
ANGELICAE ACUTILOBAE RADIX	The roots of Angelica acutiloba Kitagawa or Angelica acutiloba Kitagawa var. sugiyamae Hikino	4.0
ATRACTYLODIS RHIZOMA	The rhizome of Atractylodes japonica Koidzumi ex Kitamura or Atractylodes macrocephala Koidzumi (Atractylodes ovata De Candolle)	4.0
PORIA	The sclerotium of Wolfiporia cocos Ryvarden et Gilbertson (Poria cocos Wolf)	4.0
GINSENG RADIX	The roots of Panax ginseng C. A. Meyer (Panax schinseng Nees)	3.0
CINNAMOMI CORTEX	The bark of Cinnamomum cassia Blume	2.5
POLYGALAE RADIX	The roots of Polygala tenuifolia Willdenow	2.0
PAEONIAE RADIX	The roots of Paeonia lactiflora Pallas	2.0
CITRI UNSHIU PERICARPIUM	The pericarp of Citrus unshiu Markowicz or Citrus reticulata Blanco	2.0
ASTRAGALI RADIX	The roots of Astragalus membranaceus Bunge or Astragalus mongholicus Bunge	1.5
GLYCYRRHIZAE RADIX	The roots and stolons of Glycyrrhiza uralensis Fischer or Glycyrrhiza glabra Linn´e	1.0
SCHISANDRAE FRUCTUS	The fruits of Schisandra chinensis Baillon	1.0

## Data Availability

The data and raw materials presented in this report are available from the corresponding authors upon request.

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
