# Peer review of "Analysis of the Effects of Ninjin’yoeito on Physical Frailty in Mice"

_ijms, 2022, doi:10.3390/ijms231911183_

Round 1
Reviewer 1 Report
Dear Authors:
Thank you. Recommendations as the files.

Author Response
Responses to the comments of Reviewer 1
Comment 1: Pharmacokinetic study of Ninjin'yoeito: Absorption and brain distribution of Ninjin'yoeito ingredients in mice. Ninjin'yoeito is traditional Japanese medicine. What is the Chinese name?
Response: The Chinese name for ninjin’yoeito is ren-shen-yang-rong-tang.
Comment 2: Abstract needs to be formal. Introduction, Methods, Results, Conclusion. Results are not clear.
Response: Thank you for this comment. The abstract has been revised accordingly (Lines 23–39).
Comment 3: journal formal is Introduction, Materials and Methods, Results, Discussion, and Conclusion. Needs an introduction, materials and methods, results, discussion, and conclusion. It differs from a review article or a short report. Also different from a case report.
Response: Thank you for this comment. The paper is structured this way because it follows the MDPI submission rules and body template.
Comment 4: 8-OHDG
Response: Thank you for this comment. The introduction has been revised accordingly (Lines 72–74).
Comment 5: “The SAMP8 mice used in this study have 106 been used extensively in aging research, especially in sarcopenia studies; the IL-10KO and Sod1KO mouse models of frailty have not been used extensively in sarcopenia studies. Therefore, we hypothesized that SAMP8 would develop physical frailty based on sarcopenia, enabling us to examine the effects of NYT on physical frailty.” Showed in the results. But we did not find them.
Response: Thank you for this comment. The introduction has been revised accordingly (Lines 115–117). Furthermore, the fact that SAMP8 is physically frail based on sarcopenia is noted in the Results section (Lines 126–180).
Comment 6: Hypothesized, AIM, and Objective are needed in the first paragraph.
Response: Thank you for this comment. In this paper, we thought it would be easier to understand by stating the objectives in the second half of the Introduction.
Comment 7: Write more about ninjin’yoeito (NYT) and Oxidative stress, caspase 3 relations.
Response: Thank you for this comment. Accordingly, additional information has been included in the Introduction (Lines 90–95).
Comment 8: Histology and Immunohistochemistry, Euthanasia, Behavioral Test are not found results.
Response: Thank you for this comment. Accordingly, additional information has been included in the Results: Euthanasia (Lines 128–129), Behavioral Test (Lines 136–140, Lines 159-163), and Histology and Immunohistochemistry (Lines 166–228). Figures 1, 3, and 4 have also been corrected.
Comment 9: Results, discussions, Figures 1, 2. N=8, why Figure 3, 4. N= 5? Figure 1, no mean and SEM. This is in the tables.
Response: Thank you for this comment. The N values differ because the experiment encountered problems during the process of tissue collection and sample preparation, and the samples could not be analyzed. Accordingly, additional information has been included in the Results (Lines 136–140, Lines 159–163).
Comment 10: Discussion, Don’t talk about methods.
Response: Thank you for this comment. Accordingly, the discussion has been revised and additional descriptions have been included in the Methods section (Lines 422–425).
Reviewer 2 Report
Please explain the following points:
(1) Please comment on the animal’s well-being, did you monitor them for stress/discomfort within the cage?
(2) Please describe the dose and associated dose dependence of this experimental sample. And the results and precautions of use in humans.
(3) The authors describe that ninjin’yoeito may improve physical frailty through reductions of oxidative stress in both muscle and brain tissues. However, there is less data on antioxidants. Please provide more information about its pharmacological evidence and mechanism.
Author Response
Responses to the comments of Reviewer 2
Comment 1: Please comment on the animal’s well-being, did you monitor them for stress/discomfort within the cage?
Response: Thank you for this comment. The experimental animals used in this study were weighed daily throughout the experimental period, and stress was evaluated by observing for significant weight loss or injury.
Comment 2: Please describe the dose and associated dose dependence of this experimental sample. And the results and precautions of use in humans.
Response: Thank you for this comment. In vitro studies using isolated rat hepatocytes have reported a volume-dependent increase in the antioxidant effect of NYT; therefore, in the present experiments, a larger volume of NYT may provide better results. Regarding the use of NYT in humans, we believe that it is necessary to carefully consider the use of NYT in Japan because there is a rule of 9 g per day in Japan, and increasing the intake may cause side effects. Your comments are important and we have revised the text and added text accordingly (lines 352–355).
Comment 3: The authors describe that ninjin’yoeito may improve physical frailty through reductions of oxidative stress in both muscle and brain tissues. However, there is less data on antioxidants. Please provide more information about its pharmacological evidence and mechanism.
Response: Thank you for this comment. In this study, we focused on the antioxidant effect of NYT, but the detailed mechanism needs to be investigated in future work. We have added text to the Discussion section regarding possible considerations from the results of this study (lines 319–327). In addition, points that cannot be discussed regarding the effects of NYT are noted in the Limitations section (lines 345–347).
Round 2
Reviewer 1 Report
Dear Authors:
Your manuscript needs to write formally: 1. introduction, 2. materials and methods, 3. results, 4. discussions, and 5. conclusion.
Cancel all email addresses on the first page. On the see the corresponding authors' email addresses.
This manuscript's authors are too complex.
But....